# Compromise or choose: shared movement decisions in wild vulturine guineafowl

Danai Papageorgiou [1,2,3,4,5✉], Brendah Nyaguthii[6,7,8] & Damien R. Farine [1,2,8,9✉]

Shared-decision making is beneficial for the maintenance of group-living. However, little is known about whether consensus decision-making follows similar processes across different species. Addressing this question requires robust quantification of how individuals move relative to each other. Here we use high-resolution GPS-tracking of two vulturine guineafowl (*Acryllium vulturinum*) groups to test the predictions from a classic theoretical model of collective motion. We show that, in both groups, all individuals can successfully initiate directional movements, although males are more likely to be followed than females. When multiple group members initiate simultaneously, follower decisions depend on directional agreement, with followers compromising directions if the difference between them is small or choosing the majority direction if the difference is large. By aligning with model predictions and replicating the findings of a previous field study on olive baboons (*Papio anubis*), our results suggest that a common process governs collective decision-making in moving animal groups.

[1] University of Zurich, Department of Evolutionary Biology and Environmental Studies, Winterthurerstrasse 190, 8057 Zurich, Switzerland. [2] Max Planck Institute of Animal Behavior, Department of Collective Behavior, Universitätsstraße 10, Konstanz 78457, Germany. [3] University of Konstanz, Department of Biology, Universitätsstraße 10, Konstanz 78457, Germany. [4] Kenya Wildlife Service, P.O. Box 40241-001000 Nairobi, Kenya. [5] Wissenschaftskolleg zu Berlin, College for Life Sciences, Wallotstrasse 19, Berlin 14193, Germany. [6] University of Eldoret, School of Natural Resource Management, Department of Wildlife, 1125-30100 Eldoret, Kenya. [7] Mpala Research Centre, P.O. Box 92 Nanyuki 10400, Kenya. [8] National Museums of Kenya, Department of Ornithology, P.O. Box 40658-001000 Nairobi, Kenya. [9] Australian National University, Division of Ecology and Evolution, Research School of Biology, 46 Sullivans Creek Road, Canberra, ACT 2600, Australia. ✉email: danpapag@gmail.com; damien.farine@ieu.uzh.ch

The contribution of multiple individuals to group decision-making can bring substantial benefits[1]. Shared decisions can be more accurate[2,3], for example homing pigeons (*Columbia livia*) have more direct homing routes when flying in dyads than when flying alone[4]. Shared decisions also allow all group members to acquire vital resources while remaining part of the group[5], as they allow individuals in a state of need to influence group decisions[6]. While many studies have found evidence in support for shared decision-making, for example by observing a range of different individuals initiating movements[7–9], the extent to which collective decision-making is governed by similar movement rules across species requires further investigation[10,11].

Collective decisions can be an emergent outcome of the movement interactions among individuals[12]. The classic theoretical model that proposes this hypothesis provides two sets of testable predictions: (i) that the geometry of a conflict in preferences among initiators (the angle of their directional vectors) should determine the actions of followers, and (ii) that individuals should follow a majority rule when choosing which direction to follow[1,11,12]. The first prediction is that when faced with differences in the preferred direction of movement of group members, followers should average between directions if the disagreement among initiators is small (i.e. 'compromise') or choose one option over the other (i.e. 'choose') if the disagreement is large (above a critical angle[12]). Greater disagreement (e.g. a larger angle between initiators and/or having more initiators proposing different directions) should also reduce the probability of following[13]. The second prediction is that when choosing a direction, followers should move where the majority of preferences are directed[12]. These key predictions allow quantitative comparisons of the processes driving collective decisions across different species. However, testing these predictions is challenging, as they require information about how potential decision-makers—both initiators and followers—move relative to one-another[14].

Two studies have provided evidence for the geometric prediction of the aforementioned classic model of collective motion in semi-wild or wild animal groups[4,13]. GPS-tracking of pairs of homing pigeons showed that if the disagreement between the two birds' directional preferences when flying back home was small, individuals averaged their routes. Instead, if disagreement was over a critical threshold, either the dyad split or one of the two birds became the leader[4]. However, as the study was conducted on dyads, there was no test of the classic model's prediction on which direction individuals would choose when faced with large disagreement and a numerical difference between the clusters of concurrent initiators. That gap was covered by a study[13] that fitted GPS trackers to the majority of individuals in a troop of olive baboons, a species in which individuals form groups with very stable membership. By analysing the relative movements of individuals, and extracting initiations and following behaviours, the study showed support for the two sets of predictions for shared decision-making emerging from interactions among individuals. First, individuals were less likely to follow when there was greatest directional conflict among initiators, but when following, individual baboons averaged proposed directions by initiators when the disagreement was small and chose one or the other when the disagreement was large[13]. Second, when choosing a direction, individual baboons used a majority rule—moving in the direction with the largest number of initiators[13]. Thus, evidence is beginning to suggest that emergent decision-making processes might be relatively common across animals that move as groups, and could potentially be underpinned by a consistent set of individual decision rules.

One challenge with determining whether species use similar rules when making decisions is that careful replication is required. While the replication crisis in biology[15–17] largely stems from the

incentive structures favouring novelty[18], there are also logistical barriers to replication. For example, one recent study[19] testing whether the increase of $CO_2$ in the ocean impacts the behaviour of coral reef fish replicated previous experiments by examining a large number of captive fish (900) from multiple species (6) and across several years (3), matching the conditions of older experiments and finding low support for the original results. However, critics—right or wrongly—noted that methodological differences could also contribute differences in the results[20], meaning that the true answer remains largely unknown. The challenges that are inherent with working with whole organisms, and with the different ecological conditions that they might experience in different studies, means that replications remain relatively rare. While large-scale collaborative networks[21,22] can overcome some of the barriers to making comparative studies, large-scale and long-term studies conducted in the wild often cannot be replicated, despite these being among the most influential[23–26]. A consequence of this is not only a lack of certainty in our scientific results, but also a lack of data on the generality of our findings.

Here, we conduct a within- and between-species replication study to investigate how consensus is achieved when individuals are faced with conflicting directional preferences among group members. We study vulturine guineafowl, a sympatric species to olive baboons that also forms stable and cohesive groups. A previous observational study on collective departures suggested that every member of a vulturine guineafowl group can initiate movement from a scattered food resource but that individuals excluded from clumped food patches are more likely to lead their group after receiving aggression[6]. These findings indicated that dominance can play a role in modulating leadership—at least in the context of departures from food patches. In the present study, we fit high-resolution solar-powered GPS-trackers to almost all adults from two groups of vulturine guineafowl and implement the same analytical procedure as the previous study on baboons[13] to determine how group members reach consensus across a broader range of movements. We first confirm that all group members can successfully initiate movement, and that males (who are on top of the dominance hierarchy[27]) have greater influence on group movements than females. We then show that vulturine guineafowl express the same geometric properties and majority rule as predicted by the classic theoretical model of leadership and collective decision-making[12], and match almost exactly the empirical results observed in wild olive baboons[13]. Our study provides a powerful replication of previous empirical work, enabling quantitative comparisons between observational and GPS-based methods, and between two taxonomically distant species that live in the same habitat.

## Results

**Sex but not dominance determine who has influence**. Decisions by animal groups can be despotic, where one individual decides[28], partially shared (or graded), where some individuals contribute to decisions more than others[29], or fully shared, where all individuals have an equal influence[13]. When contribution to decision-making is unequal, it has generally been predicted that dominant individuals should have greater influence[30], but this has received mixed support[28,31–34].

We first explored whether leadership in vulturine guineafowl is fully shared or graded by quantifying the role of dominance on influence. Vulturine guineafowl groups have steep dominance hierarchies (see Supplementary Fig. 1), which remain stable for several months[6,27]. We applied the approach described by Strandburg-Peshkin et al.[13] to infer who initiates movements and who follows, based on dyadic movement patterns from the

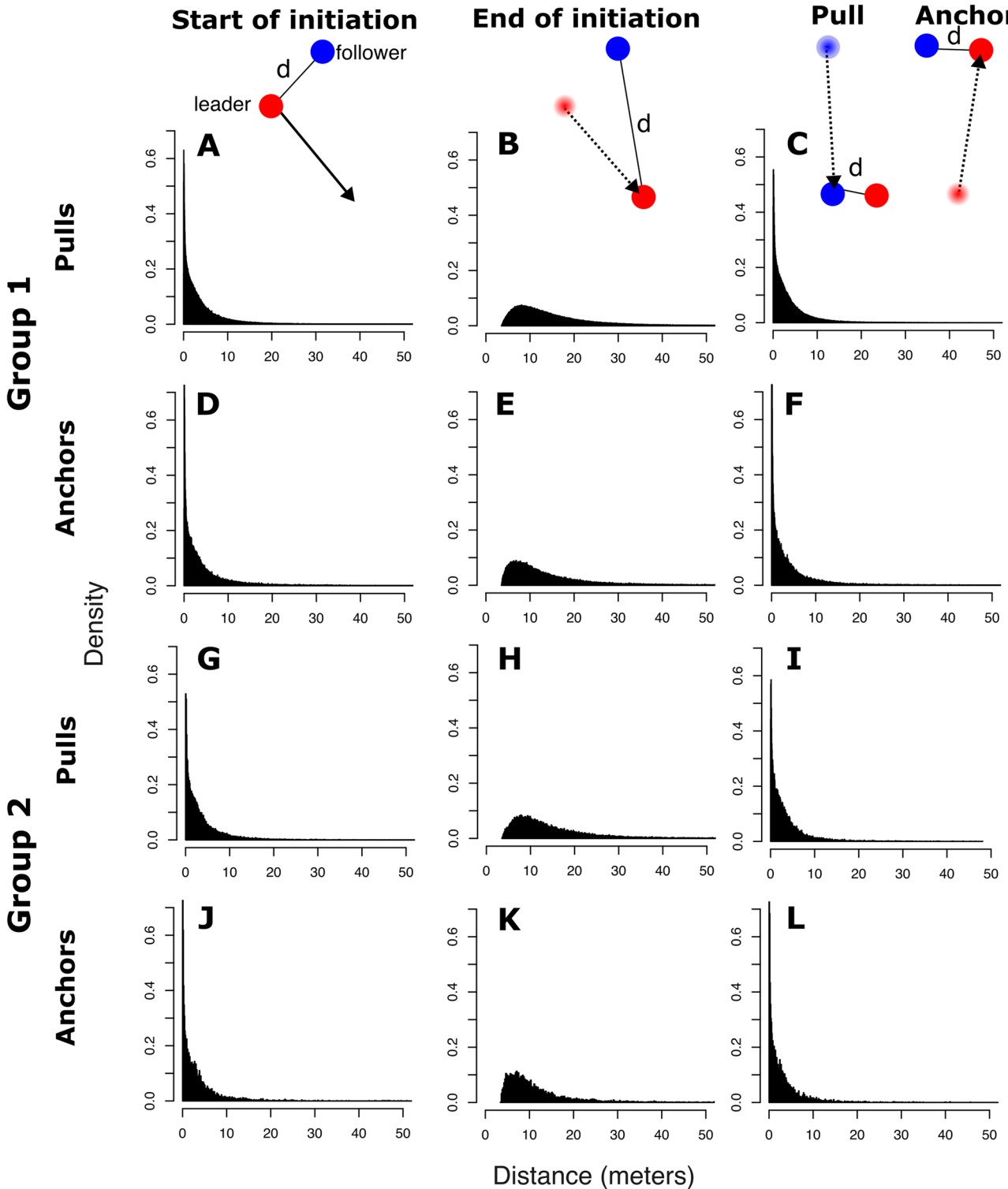

**Fig. 1 The distribution of distances (d, in metres) between initiators and potential followers, during pulls and anchors at three different times.** First, when the initiation starts (first column; **A**, **D**, **G**, **J**); second, when the initiator reaches the maximum distance away from the potential follower (second column; **B**, **E**, **H**, **K**); and third, when the potential follower starts moving towards the leader (in pulls; **C**, **I**) or the leader returns by moving towards the potential follower (in anchors; **F**, **L**). Distributions of inter-individual distances are shown separately for groups 1 (**A**–**F**) and 2 (**G**–**L**).

GPS tracks collected simultaneously across group members (see Methods for details on GPS tracking). Initiation attempts were characterised by an increasing inter-individual distance followed by a decreasing inter-individual distance (see Methods and Fig. 1). Depending on the relative contribution of each individual to the change in distance, initiations were classified as being successful

('pulls', where A moves to increase the distance and B moves to subsequently reduce the distance) or unsuccessful ('anchors', where A moves to increase the distance but, subsequently, decreases it by moving back towards B).

Summarising 502,253 leader-follower cases from two social groups, we confirm that all group members can initiate movement

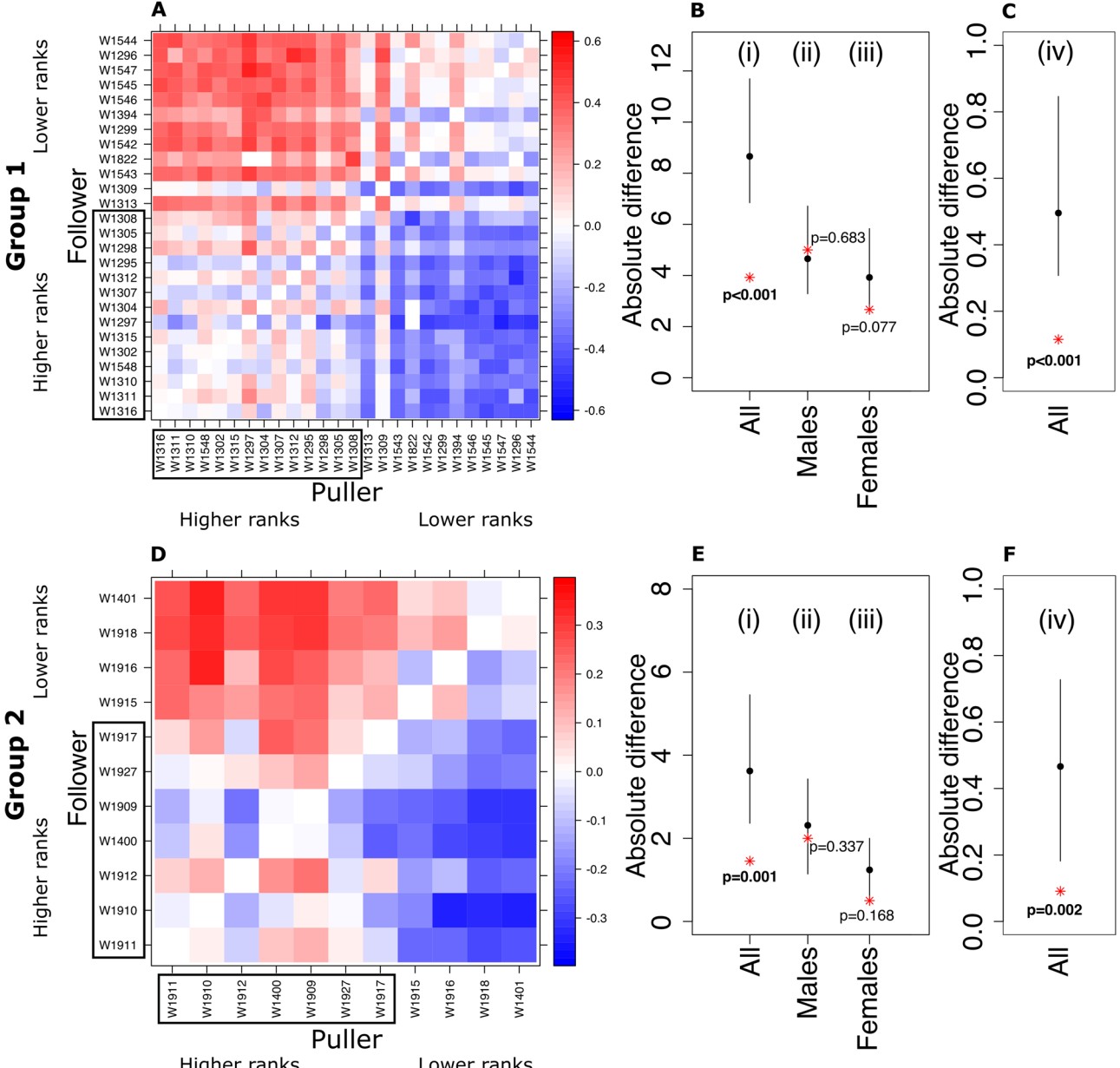

**Fig. 2 Dyadic influence, and relationships with dominance and sex, across two social groups. A**, **D** In each group, we calculated the dyadic influence index, ranging from 1 (red) when the individual in the column always leads the individual in the row to −1 (blue) when the relationship is reversed. Individuals are plotted in descending dominance rank, e.g. the alpha male of Group 1 is "W1316" and the lowest ranking group member being the female "W1544". Individuals in rectangles are males. **B**, **E** The result of permutation tests on the relationship between dominance and leadership across all individuals (i), within males (ii), and within females (iii). While there appears to be an effect of dominance on individual influence (i), this effect is not present within sexes (ii-iii). The y-axis corresponds to the absolute difference between dominance rank and influence rank (see Methods). **C**, **F** A permutation test on the relationship between sex and influence confirms that the relationship is driven by males being more influential than females. The y-axis shows the absolute difference between two binary variables; whether an individual was within the top n-ranked individuals, where n represented the number of males in the group and whether the individual was a male or not (see Methods). Panels **B**, **C**, **E**, **F** show the observed value (red star) relative to the mean (black dot) and 95% range from 1000 permutations of the datasets. P values are calculated as the proportion of the number of values of the randomised data being larger than the observed values divided by the number of randomised values (i.e. 1000).

and pull others, but that there is a distinct subset of individuals that are more likely to be followed (Fig. 2A, D). To investigate the relationship between dominance and the probability of being followed, we ran permutation tests within and between sexes (Fig. 2B, C, E, F, see the Methods section for details on the permutations). While it appears that more dominant individuals are more likely to be followed (Fig. 2B i, E i), analyses controlling for sex show it was rather that males, who are dominant over

females, are more likely to be followed (Fig. 2B ii-iii, C, E ii-iii, F), and that there is no effect of dominance within sex. In a two-puller context comprising one male and one female initiator and where followers choose one direction (see below), the effect of sex translates to a difference in success rate of approximately 10% (Group 1; $P_{male\ success} = 0.543$, Group 2; $P_{male\ success} = 0.553$). However, success is not only determined by the probability of being followed, but also by the rate of initiating. When considering the

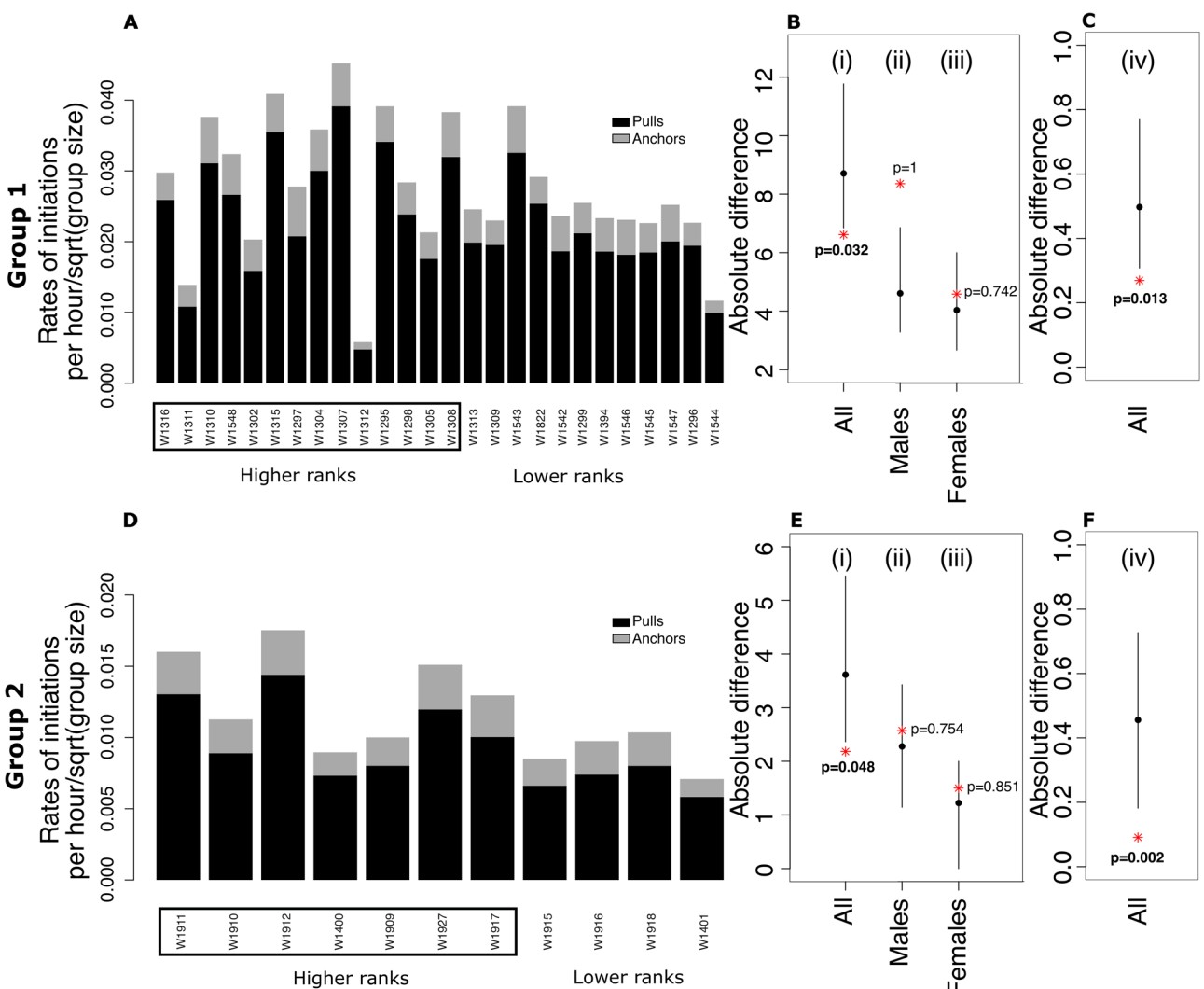

**Fig. 3 Initiation rates, and relationships with dominance and sex, across two social groups.** Panels **A** and **D** show the number of times each individual pulled another group member (black) or was anchored (grey). Individuals are plotted in descending dominance rank and those in rectangles are males. **B**, **E** The result of permutation tests on the relationship between dominance and successful initiation rates per hour (i.e. pulls) across all individuals (i), within males (ii), and within females (iii). While there appears to be a weak effect of dominance (i), this effect is not present within sexes (ii-iii). **C**, **F** A permutation test on the relationship between sex and successful initiation rates per hour (i.e. pulls) confirms that the relationship is driven by males initiating more often than females. The y-axis shows the absolute difference between two binary variables; whether an individual was within the top n-ranked individuals, where n represented the number of males in the group and whether the individual was a male or not (see Methods). The y-axes in **B**, **C**, **E**, **F** correspond to the absolute difference between dominance rank and ranked values of initiation rates. Permutation tests and panels are as per Fig. 2.

number of successful initiations for each individual, we again find no effect of dominance but a consistent effect of sex (Fig. 3).

Our results show that leadership in vulturine guineafowl is shared, aligning with the previous work on olive baboons[13] and with direct observations in this system[6]. However, unlike in baboons, leadership in vulturine guineafowl is not completely equal. Instead, it is graded, with males being more likely to be followed and initiating at higher rates (on average) relative to females. This difference to baboons may relate to the fact that males, who are dominant, are also the philopatric sex in vulturine guineafowl[35]. Staying in their natal group potentially allows guineafowl males to maintain life-long, and thus stronger, influence relationships with other members of their group. The natal sex also has more information about the local landscape than the dispersing sex, which may contribute to observed differences (though we note that the baboon study[13] did not explicitly tested for a sex difference). This difference may not,

however, always play a role in decision-making. Females were often as successful at initiating as male group members, and many females initiated movements more often than some males. The relatively small differences between male and female vulturine guineafowl is likely to reflect the relatively low rates of conflict in most of their collective movements. We found that guineafowl are substantially more likely to follow initiators ($P_{success}$ range: 0.7–0.9) than baboons ($P_{success}$ range: 0.2–0.8)[13]. This is likely to explain the high degree of cohesion and small intra-group dispersion of vulturine guineafowl.

**Individuals are more likely to follow when initiators agree.** We aggregated the simultaneous initiation attempts acting on single candidate follower individuals into 'events' based on their over-lapping start and finish timestamps, following Strandburg-Peshkin et al.[13]. While initiation attempts lasted on average for

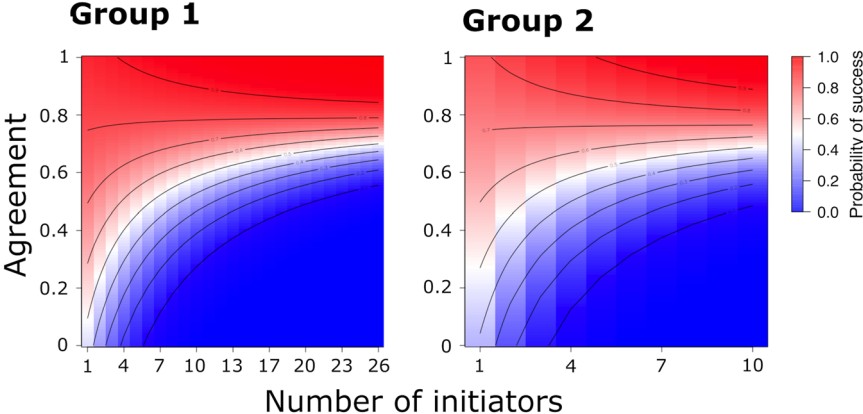

**Fig. 4 An individual is more likely to follow when there are few simultaneous initiators, and, as the number of concurrent initiators increases, when initiators have high agreement.** As the number of initiators increases, then the directional agreement becomes increasingly important in determining the decision of a follower, as revealed by the positive interaction term in the GEE models (Supplementary Table 1).

2.4 min (SD = 2.7), the temporal overlapping nature of initiations that were combined into events meant that events lasted longer than the initiation attempts themselves, with an average of 7.0 min (SD = 6.2; Supplementary Fig. 2). For each event, we calculated the direction of the initiators in relation to the position of the potential follower and their directional agreement. Directional agreement ranged from 0, when the movement vectors of initiators were equally distributed over potential directions, to 1 when the movement vectors were perfectly aligned (see Methods section for more details). We also noted whether the potential follower was subsequently pulled or not, and calculated the direction of the movement of the follower if the follower was pulled. We defined events as successful if at least one initiator pulled the potential follower.

We found that the number of simultaneous initiators, the level of their directional agreement, and the interaction between these two, all predict the probability of following a given initiation. In both groups, increasing the number of initiators has a positive effect on following when the angular agreement was high, but a negative effect on following when the agreement was low (Fig. 4; Supplementary Table 1). Although the results are consistent across both groups, only the interaction is significant in Group 2 (for which we collected substantially fewer data; see Supplementary Table 2). Supplementary analyses that account for changes in the number of tracked individuals and the distance between initiators and potential followers confirm that our results are robust to variation in data collection and to the assumptions of the methods.

The interaction between agreement and the number of initiators on the tendency for vulturine guineafowl to follow, matches closely with the behaviour of baboons. Specifically, baboons also require greater agreement when there are more initiators in order to follow[13]. In vulturine guineafowl, the patterns are also very similar across both groups: having more simultaneous initiators requires a higher agreement for individuals to follow, and high levels of agreement (>0.6) generally result in a better-than-chance (>0.5) probability of an event being successful. Baboons appear to be more tolerant of disagreement, with any agreement over 0.3 producing a better-than-chance probability of an initiation being successful[13].

**Followers compromise the initiation directions when initiators agree but choose a direction when initiators disagree**. For each successful event, we tested the theoretical prediction[12] that the angular agreement of the initiators should determine where a follower moves next. For simplicity, in this particular test we

focused on events comprising two initiators (17.160% of all events for Group 1 and 18.157% of all events for Group 2, see Supplementary Table 2), allowing us to calculate the angle between the initiators relative to the potential follower. If a follower moves in a direction that averaged the angle between initiators (i.e. 'compromise'), then we expected a unimodal distribution in the directions taken by followers across repeated observations at a given angular disagreement. By contrast, if a follower 'chooses' one or the other direction, then we expected a bimodal distribution in the directions taken by followers across repeated observations with the same angle of disagreement.

We found that the direction taken by vulturine guineafowl followers has identical properties to those predicted by theory and those found in baboons. Specifically, in both guineafowl groups, followers compromise the initiated directions when the disagreement between initiators is below a critical threshold that separates the two regimes, and choose one direction versus the other when the disagreement is above the threshold (Fig. 5, Supplementary Fig. 3).

As with previous results, we found strong concurrence between vulturine guineafowl and baboons. Baboons also express a transitional phase from compromise to choose, which is estimated to range between 72 and 96 degrees[13]. In Group 1, we found that the lower end of the transitional phase from compromise to choose is almost identical to that of baboons (78 degrees), but that the upper end is much higher (130 degrees). In Group 2, we could only find a transition threshold, which is estimated to be 117 degrees. But, as estimates for Group 2 are based on substantially fewer data, we expect that adding more data will reveal a larger range of uncertainty as in Group 1. The data from Group 2 do, however, also suggest that the upper end of the transition phase to the choose regime takes place at a larger angle in vulturine guineafowl than in baboons.

**Followers move in the direction of the majority when choosing**. To find where a follower moves when in the choose regime, we focused on cases when two or more individuals initiated toward different directions. We used a spatial clustering algorithm to identify sets of individuals co-initiating in similar directions, extracted cases in which there were exactly two clusters, and counted the number of individuals initiating in each of the two directions.

As predicted, we found support that vulturine guineafowl employ a majority rule when choosing one versus the other direction to move in (i.e. at high levels of disagreement, on the right side of each panel of Fig. 5). Specifically, in both Group 1

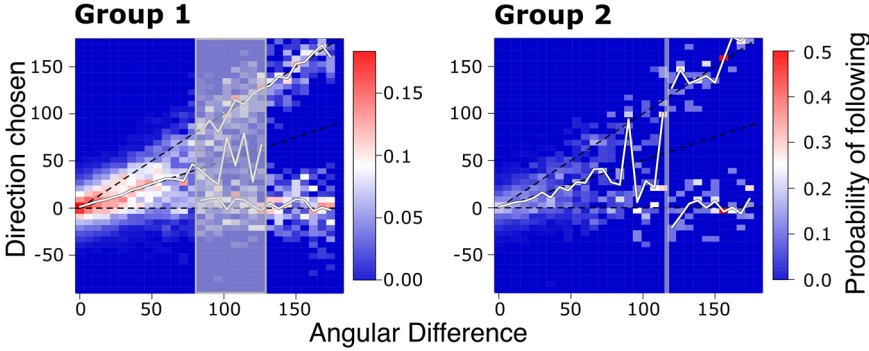

**Fig. 5 When following, vulturine guineafowl transition from compromise to choose depending on the angular disagreement between initiators.** Plots show the angle between two initiators relative to a potential follower (x-axis) and the resulting direction taken by the follower (y-axis). When the angular difference between initiators is above a critical threshold (see grey rectangle for transitional zone), follower directions are significantly bimodal (see Supplementary Fig. 3), suggesting that followers choose one direction or the other. Colours blue, white and red show the probability of a direction to be chosen by a follower. Solid white lines represent the median of the chosen direction(s) under the compromise and choose regimes (or both in the transitional zone).

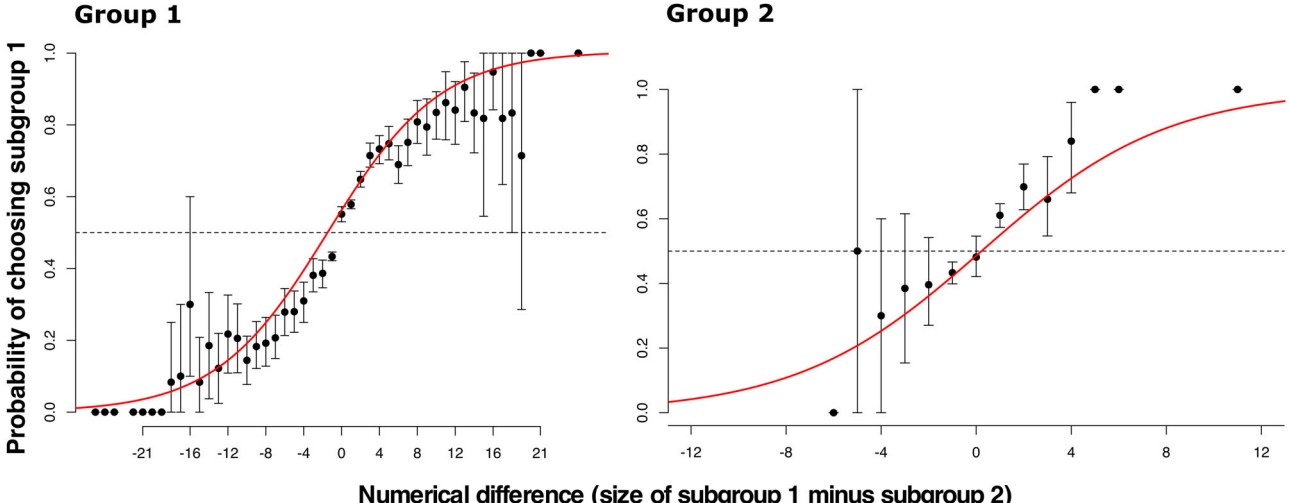

**Fig. 6 When two clusters of initiators propose different directions with a large angle of disagreement, followers disproportionately choose the direction of the largest cluster of initiators, thus following a majority rule.** Black dots represent the empirical data and error bars are 95% confidence intervals estimated by 1000 bootstrapped replications of the data. The red line shows a significant sigmoidal fit to the data. Model summaries are presented in Supplementary Table 3.

and Group 2, followers are disproportionately more likely to move in the direction containing the largest cluster of initiators (Fig. 6, Supplementary Table 3).

Our results confirm that vulturine guineafowl use a similar majority rule to baboons when choosing between directions. In baboons, individuals have an 80% chance of choosing the majority when the difference between the number of initiators in each cluster is three or more. By contrast, the model fits predict that vulturine guineafowl require a larger numerical difference (a difference of 8–9 for Group 1 and 4–5 for Group 2) to reach the same level of discrimination.

## Discussion

Our study shows that the movements of vulturine guineafowl are consistent with the predictions from a classic theoretical model of leadership and collective decision-making, and have striking similarities to the movements described in taxonomically distant but sympatric olive baboons[13]. In both our guineafowl study groups, we found that any individual could initiate movement, with no direct link between dominance and influence. Male guineafowl

are more likely to be followed than females, and also have slightly higher rates of initiations. However, females still initiated often, and many had a high number of successful attempts. Like in baboons, conflicts in vulturine guineafowl group decisions affect the probability that initiators are followed and, when they do, follower movements fall into one of two regimes: when the disagreement between concurrent initiators is small followers average the directions of the initiators and when the disagreement is large they choose the direction with the most initiators. Our study also demonstrates the importance of replication in ecology and animal behaviour[15,16,36], showing that by following the same methods and conducting the same statistical tests we could reveal that the emergence of collective decisions from simple rules governing group cohesion are likely to be consistent across very distinct taxonomic groups.

While influence can be distributed within the group[37], whereby all individuals can initiate movement and be followed, it is not necessarily equal among group members[29,38,39]. For example, homing pigeons form influence hierarchies during flight and these hierarchies determine whom an individual is likely to lead and are most likely to be led by Nagy et al.[32]. However, these influence

hierarchies are independent from dominance hierarchies[33]. In vulturine guineafowl, we found that leadership is generally shared, but that males are more likely to be followed and initiate more often than females. This difference reflects the social structure of vulturine guineafowl societies, where males are dominant over all females[6,27], who are also the dispersing sex[35]. However, males have been found to be more influential than females in collective departures also in species in which female matrilines dominate aggression hierarchies[7,40], suggesting that neither the dominance hierarchy alone, nor the dispersal tendency of the sexes, always determine which individuals influence group coordination. Further, while the differences we found between males and females are significant, females still had substantial influence over where groups moved. One key outstanding question is therefore to identify whether there are specific contexts in which the ability to exert influence (i.e. have a higher probability of being followed) may be important.

Within each sex group of vulturine guineafowl, we found no evidence for dominance playing a role on influence (although in both groups, the lowest ranking female initiated less often than almost any other group member). While it appears that female vulturine guineafowl have overall less influence than males on where their group goes, unlike in other species[41–45], it is also possible that females exhibit specific strategies to influence decisions[46,47]. For example, they could influence when groups leave[6], as has been suggested in baboons[48], with timing decisions potentially reflecting a distinct axis of decision-making[13,49]. Such routes of influence would not be obvious from the analytical approach that was employed in the current and baboon studies[13]. Rather, our approach likely captured a number of very general collective movements, including many moment-by-moment decisions (e.g. which way to move around a tree). Identifying the functional importance of each decision (e.g. those that dictate where groups move next at larger spatial scales) remains a challenge in the field.

Despite the properties of follower movements in response to initiators being very similar across vulturine guineafowl and baboons, details may vary from group to group[50] and from one context to the next[51]. In theory, larger group sizes should be associated with a decrease in the angle at which follower movements transition from compromise to choose[12]. Group 1 was similar in size to the previously studied baboon group, and their transitional phase overlapped[13]. Further, the transitional phase of Group 1 started at values that were almost 40 degrees smaller than the smaller Group 2, albeit it also went beyond that of Group 2 (Fig. 6). Given this overlap, we can't safely draw conclusions on whether our findings support the theoretical predictions that indicate that larger groups show a smaller transitional angle, and therefore data on more groups of different sizes are required to address this question. The potential influence of the social, as well as the physical environment is worth exploring, given that environmental effects have already been documented across various facets of collective behaviour[51].

One behaviour where we could find some clearer differences between groups is when looking at the majority rule employed by each. In vulturine guineafowl, the smaller group (Group 2) appeared to require a smaller threshold in order to identify a clear majority. A key question is whether the shift in the threshold scales with group size. Our data suggest that the larger group (Group 1) reliably chose the majority (80% of the time) when there was a higher proportional difference (approximately one third of the group 1) in initiators compared to the smaller group (approximately one quarter of the group 2). Baboons required a much smaller majority to reach the same 80% threshold[13]. Given that the baboon troop that was studied was larger than Group 1 (and had a very similar proportion of GPS-tracked group members), these

results suggest that discrimination may be harder in larger groups, and that baboons could have a better capacity to discriminate smaller relative differences than vulturine guineafowl.

Our results show that the processes driving the movement patterns of wild group-living vulturine guineafowl are largely consistent with those previously described in a group of wild baboons[13] and in dyads of homing pigeons[4], with the specific directions of movements by individuals and responses to conflict when following in particular matching those of olive baboons. Our work adds to the weight of support for predictions arising from a classic theoretical model of leadership and collective decision-making[12]. Further, by carefully conducting a large-scale within- and between-species replication, we propose that a multitude of group-living species could exhibit highly convergent processes governing how they reach consensus on where to move.

## Methods

**Data collection.** Our study population of vulturine guineafowl resides in a savannah-woodland ecosystem of approximately 12 km[2] in the southern part of the Mpala Research Centre (MRC) in Laikipia, Kenya. Vulturine guineafowls are large (~1.5 kg), predominantly terrestrial, and live in relatively large groups (13–65 adults) with largely stable membership[52]. Groups are not territorial and associate preferentially with specific other groups[52].

*GPS trackers.* We fitted with GPS solar-powered tags almost all adult members from two groups of vulturine guineafowl. We programmed the GPS tags to simultaneously collect 1 Hz data every fourth day from 06:00 to 19:00 by allowing them to fully recharge over three days before starting a full day of operating. For the purposes of other research projects running at the same time[53], we set one to two tags in each group to work on a daily schedule and during some months the tags of all individuals in focal Group 2 where programmed to work on a daily basis (see Supplementary Table 4 for the group size per month, number of tagged individuals, how long they were tracked and GPS tag programming setting, see also Supplementary Movies 1–6 for a demonstration of the whole-group tracking datasets of Groups 1 and 2). This 'daily' setting recorded one data point (date, time, coordinates) every second when the battery had a high charge (approximately every second to third day, for up to 8 h continuously). When the battery was at the next highest threshold, tags recorded 10 points spanning the first 10 s of every fifth minute. At the lowest battery threshold, tags recorded one point every 15 min (this setting was used less than 1% of the time). We downloaded data remotely every two to three days using a BaseStation II (e-obs Digital Telemetry, Grünwald, Germany).

We conducted census observations every two days (on average) to record changes on group size and the number of tagged individuals per group across the study period, as some individuals got predated or lost their tags. We summarise this information in Supplementary Table 4.

*Dominance hierarchies.* First, to estimate the dominance hierarchy, we conducted all-occurrence sampling in each group, recording different types of agonistic interactions, as described by Papageorgiou & Farine[6] and Dehnen et al. [27]. For each observed interaction, we recorded the time, the winner, and the loser. We recorded data over at least 3 sessions, lasting 2-3 h each, per group, per week across the study period (restricted to days when simultaneous GPS tracking was not taking place). From the agonistic interactions data, we calculated a dominance hierarchy for each group using the randomised Elo scores method[54].

To test if the dominance hierarchy remained stable during the study period, we calculated the repeatability score of ranks by

randomising the order of the data, splitting the dataset in two halves, and calculating the Spearman rank correlation coefficients across the estimates of ranks from each half. We repeated this process 1000 times, using the function 'estimate_uncertainty_by_splitting' from the 'aniDom' R package[54], to estimate a mean and 95% confidence intervals of the correlation values.

**Data processing**. We used (and adapted where necessary) the methods and published code developed by Strandburg-Peshkin et al.[13]. We repeated each of the following steps on the data from the two study groups separately.

*Pre-processing GPS data.* We used the built-in features from the Movebank data repository to remove the outliers from our dataset that were falling outside of our study area (<0.001% of the data, corresponding to points that were often outside of Kenya). In the rare cases when a tag failed to log one point (e.g. skipping one second, 0.16–0.21% of the data in both groups), we linearly interpolated missing points based on the existing data around that point from the same tag. More specifically, if there was a missing value at time t, between t − 1 s and t + 1 s, we added one point in time t, in the middle of the straight line connecting the two known points of t − 1 s and t + 1 s.

*Extracting successful and failed initiation attempts at the dyadic level.* We extracted movement initiations, and their outcomes by identifying maxima and minima in the dyadic distance between a given pair of individuals. The data between a minima and a maxima identified cases when an individual i moved away from another j (i.e. an initiation). The subsequent behaviour of individual j between the maxima and the following minima determined the interpretation of the event. If j moved towards the direction of i, the outcome was defined as a "pull", whereas if i moved back towards j, then the outcome was defined as an "anchor".

We used a set of thresholds to remove pulls and anchors potentially arising from GPS noise or small movements. Specifically, we defined initiation events as only those in which the minimum change in distance between i and j was more than 3.5 m. We believe this threshold to be biologically relevant considering the scale that the movements of vulturine guineafowl take place, especially given their high degree of spatial cohesion. It is also above the error of the GPS tags, as our field testing suggested that the estimated relative position of two GPS tags is accurate to within 1 m more than 95% of the time[52]. Further, we determined that pull or anchor events required one individual doing a disproportionate amount of movement, setting a "disparity" threshold of 0.1, whereby 0 represents both individuals having moved equally during an event and 1 represents a single individual having done all of the moving during the event. Finally, we set a "strength" threshold to 0.1, which could range from 0 when the change in dyadic distance was very small relative to the total dyadic distance (i.e. small movements by individuals far away) to 1 when the change in dyadic distance was very large compared to the total dyadic distance (large movements by individuals that are in the same spot). The latter two are the same settings as the original study by Strandburg-Peshkin et al., whereas we set the minimum change in distance to a smaller value (3.5 m instead of 5 m) as vulturine guineafowl are substantially smaller and more cohesive in their movements than baboons. The dyadic distances throughout the process of initiation are shown in Fig. 1, confirming the small distances over which leadership interactions take place in vulturine guineafowl.

We only kept in subsequent analysis events that took place when at least half of group members' tags were collecting data, which largely matched the distribution of the data in the original baboon study. In that study, 80% of adults and subadults were tagged, however some tags stopped working for periods of data collection, meaning that as few as 16 of the 26 collared baboons (55%) collected data on some days[13]. We also applied, and present, the results using a threshold keeping only events when at least 80% of group members' tags collected data at the same time. The results are presented in the Supplementary Note 1 of the Supplementary Materials (Supplementary Tables 5–8 and Supplementary Figs. 4–8) and show that the patterns in our results are not sensitive to the choice of threshold.

*Identifying simultaneous initiation events.* To investigate pulls and anchors beyond the dyadic level, we grouped together interactions (potential pulls and anchors) that operated simultaneously (i.e. involving one or more initiation attempts that overlapped in time) on one potential follower, and we defined this as an event. We considered interactions as overlapping in time using a chain rule, meaning that if interaction A overlapped with B, and interaction B overlapped with C, then all three would be combined into one event regardless of whether interaction A overlapped with C. For each event, we calculated the direction of the initiators in relation to the position of the potential follower, whether the potential follower was pulled or not, and the direction of the subsequent movement of the follower if the follower was pulled. We defined events as successful if, and only if, at least one initiator was recorded as having pulled the potential follower. To test for a majority rule—whether followers moved in the direction with most initiators, we also clustered of initiators according to their direction using Gaussian Mixture Models[55].

## Statistics and reproducibility

*Does dominance predict influence?.* To investigate if dominance predicts influence within each group, we created a matrix representing the relative influence among dyads, with the influence index in dyad, i and j defined as $I_{i,j} = \frac{P_{i,j} - P_{j,i}}{P_{i,j} + P_{j,i}}$, where $P_{i,j}$ represents the number of events individual i pulled individual j. The index ranges from −1 (j pulled in all events) to 1 (i pulled in all events), with 0 representing no difference in influence among the two individuals. From these data, we calculated influence ranks by summing each individual's indices and ranking these sums such that individuals with a larger sum were considered to be more influential.

We examined the effects of dominance and sex on influence rank by running four permutation tests:

(i) We calculated the mean absolute difference between dominance rank and influence rank for each individual. If dominant individuals were more highly ranked in the influence matrix, then we expected this value to approach 0. We evaluated the significance of our measure by recalculating the same value 1000 times after randomising the order of individuals' dominance ranks relative to their influence ranks.

(ii–iii) We tested whether there was a within-sex effect of dominance by conducting the same test as (i) in males and females independently.

(iv) We tested whether males were more likely to be on the top of the influence hierarchy by calculating the mean of the absolute difference between two binary variables. The first variable represented whether an individual was within the top n-ranked more influential individuals, where n represented the number of males in the group. The second binary variable represented whether the individual was a male or not. We then re-calculated this value in 1000 permutations randomising the link between the two binary variables.

Then, we also examined the effects of dominance and sex on the successful initiation rates (pulls) per hour. To do this, we ran permutation tests similar to (i-iv), but we replaced influence rank with the individuals ranked according to the rates of successful initiations per hour.

In all the permutation tests, we considered an effect to be significant at $\alpha = 0.05$ if the observed value was closer to 0 than 95% of the values generated by the permuted datasets.

Finally, to examine the effect of sex in the success rate of initiating, we extracted cases in which there were two simultaneous pullers comprising one male and one female initiator. We then calculated the proportion of time the male was the successful puller. We tested the significance of this measure by randomising the sexes across all of these events 1000 times. This allowed us to test if males were more successful pullers than expected by chance. As above, we considered an effect to be significant at $\alpha = 0.05$ if the observed value was smaller than the probabilities of 95% of the permuted datasets.

*How do the agreement and number of initiators affect whether initiators are successful?.* We first tested the factors that contributed to individuals' decisions about whether to follow or not. From the full set of events, we constructed a generalised estimating equations (GEE) model testing whether a focal individual would follow or not (binary response variable where pull = 1 and anchor = 0) was predicted by the level of agreement, and the number of initiators. We quantified directional agreement among simultaneous initiators using the circular variance (cv) of the unit vectors pointing from the potential follower to each initiator in the event, and defined agreement as $1 - cv$. Values of agreement are close to 0 when individuals initiate in opposing directions and approach 1 when all individuals initiate in the same direction. Given that events, include all simultaneous initiations by default, our GEE did not include an autocorrelation structure. We used the R package 'geepack'[56] to fit the GEE model.

*How does the angle between initiators affect where followers move?.* We tested whether the angle between initiators predicted where individuals moved in cases where a guineafowl did follow an initiation, focusing on events comprising two initiators. To identify which regime followers used (compromise or choose) for a given angle of disagreement, we ran a dip test of bimodality and a converging modes test (using the method developed by Hartigan & Hartigan[57], and the code from Strandburg-Peshkin et al.[13]). If vulturine guineafowl were in the compromise regime, then the distribution of angles taken by the follower would not be significantly bimodal (according to the dip test) and would be more unimodal than expected by chance (according to the converging modes test). If neither of these conditions held, then vulturine guineafowl were in the choose regime. We interpreted situations in which one condition held but not the other as demarking a transition between the compromise and choose regimes. We ran these analyses by combining events into 12 degree bins of angular disagreement. We conducted this analysis independently on both groups, using code developed by Strandburg-Peshkin et al.[13].

*Where do guineafowl move when they choose one direction versus the other?.* Finally, we investigated which direction followers chose when in the choose regime by examining the numerical difference among clusters of simultaneous pullers. Specifically, we expand the analysis in the previous section by looking at all events with more than one puller. In each of these events, we used a circular clustering algorithm[13] to identify clusters of individuals pulling in similar directions. We then extracted all of the events containing two clusters, and counted the number of individuals in each of the clusters. We then identified which of these clusters was successful, and related this to the numerical difference in the size of each cluster.

If guineafowl follow a majority rule, then they should be much more likely to follow numerically larger clusters. Following Strandburg-Peshkin et al. 2015[13], we first fit a non-linear least square model where the response variable was the probability of choosing a randomly allocated cluster 1, while the predictor was the numerical difference between the number of individuals in cluster 1 minus the number of individuals in cluster 2. We then estimated the uncertainty for each bin (i.e. for each numerical difference between the size of initiating cluster 1 minus the size of initiating cluster 2) by drawing n samples from a uniform distribution, where n is the number of events in that bin, and calculating the probability that the random values are less than or equal to the observed probability. We repeated this process 1000 times, and extracted the lower 2.5th and upper 97.5th quantile of these probabilities as a measure of the 95% confidence intervals.

**Reporting summary**. Further information on research design is available in the Nature Portfolio Reporting Summary linked to this article.

## Data availability

Processed data can be found on Figshare. https://doi.org/10.6084/m9.figshare.24850551. Raw GPS data are stored on https://www.movebank.org.

## Code availability

We used the code from Strandburg-Peshkin et al. 2015 Science. Our adjusted code can be found on Figshare. https://doi.org/10.6084/m9.figshare.24850551.

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

## Acknowledgements

We are grateful to Ariana Strandburg-Peshkin, Vivek Hari Sridhar, Margaret C. Crofoot, and Dora Biro for feedback on early versions of the manuscript as well as to five anonymous reviewers. We also thank Wismer Cherono and John Ewoi for field assistance. The research was funded by the European Research Council (ERC) under the European Union's Horizon 2020 research and innovation programme (grant agreement no. 850859 awarded to D.R.F.), the Max Planck Society, and grants awarded to D.R.F. from the Daimler und Benz Stiftung (32-03/16) and the Association for the Study of Animal Behaviour. D.P. received additional funding from a DAAD PhD fellowship and an Early Career Grant from the National Geographic Society (WW-175ER-17).

## Author contributions

D.P. and D.R.F. conceived, designed the study and performed the analysis. D.P. and B.N. collected the data. D.P. and D.R.F. drafted the manuscript. D.R.F. supervised all aspects of the study. All authors contributed to revisions.

## Funding

## Competing interests

The authors declare no competing interests.

## Ethics approval

Our study involves critical participation reflected on authorship and supervision by local (i.e. Kenyan) academics. We have complied with all relevant ethical regulations for animal use and we obtained permits from the National Science and Technology Council (NACOSTI permit: NACOSTI/P/16/3706/6465), the National Environment Management Authority (NEMA permit: NEMA/AGR/67/2017), the Kenya Wildlife Service (Research Authorizations and Capture Permits), the National Museums of Kenya (NMK). We especially thank Dr. Peter Njoroge from the Ornithological Section of NMK for providing useful feedback and reviewing our work but also the Mpala Research Centre, and the Max Planck Society's Ethikrat Committee for ethical permission to conduct this research.
