## [Peer Review File · Communications Biology]

Reviewers' comments:

Reviewer #2 (Remarks to the Author):

This manuscript is about the ways the members of the groups of terrestrial bird vulturine guineafowl (from now on birds) make compromises and choices while moving around. It focuses on several aspects arising in the above context. The authors consider the roles of such factors as dominance, the number of simultaneously acting (initiator) birds and other possible factors including, e.g., whether the collective motion is initiated by males or females. Another important aspect of their study is the comparison of their results with those determined for the collective movements of baboons and with those which are expected from a related model concerning leadership in group motion. The paper is well written, the results are interesting, however, I have a few points which I think should be clarified before the manuscript becomes suitable for publication. Before mentioning some of my concerns, I would like to point out that I agree with one of the main concepts of the work: I also think that demonstrating that the same behaviour occurs for different species is an important goal. Because of the complexity of animal societies, it can be instrumental if similarities or differences are found. In this sense I welcome "replication".

On the critical remarks side, I miss information/data regarding three relevant issues:

- 1) One of the main results (statement in the paper) is that the patterns of collective movement of the birds are analogous to those found for groups of baboons and predicted by theory. However, I do not find any quantitative comparison (figures, etc) of the present data and those which they are compared with. In other words: is the agreement they find quantitative to any degree? Or is the behaviour "similar" only?
- 2) The authors have already published several papers on these birds. It is not clear from the Methods section whether this study uses original data, or it uses data obtained earlier (for purposes involving movement interactions among the birds), and thus the present work being a re-analysis of prior data.
- 3) Since the whole work is about motion, I miss any visual demonstration of the processes described in the manuscript. I think that in 2023 a paper about the group motion of animals should have a few supplementary videos. I understand that it is kind of late to make videos of the birds (I certainly would have tried to do it, e.g., from above, using a distant, quiet drone, it takes only a few days that the birds would get completely used to it), but the authors have a huge amount of GPS coordinates versus time, so creating an animation of those would have not represented an enormous effort.

I have a few – in part minor - comments which are likely to be helpful in making the manuscript clearer about some relevant details.

- Minor: In Fig. 1 the panel (iv) is not referred to (at least I do not see what is along the x axis)
- One of the most important aspects of the phenomenon is an event called "simultaneous pull". Now, here the word simultaneous is essential and assumes a time scale which is much shorter than the typical time scale needed for a bird to make a distance compatible with the inter-bird distance. Simultaneous and "subsequent" are important aspects here and without explicitly specifying Δt , the time interval within which movements are considered to be simultaneous, the presentation of the data is incomplete.

- Minor: I find that the following sentence is so complicated that it is difficult to make a clear conclusion out of it:

"We found that the number of simultaneous initiators, the level of their directional agreement (ranging from 0 when the movement vectors have an 180° difference, to 1 when the movement vectors have

no difference), and their interaction predicted the probability of following a given initiation.”

Reviewer #3 (Remarks to the Author):

“Compromise or choose: Shared movement decisions in wild vulturine guineafowl”

Summary and overall impression of the work

This paper uses high-quality trajectory data for wild guineafowl groups. The study system, site, and data are wonderful, and it is exciting to see such analyses being undertaken. Acquiring these data takes a huge effort and dedication – well done to the authors. The framing of the study and discussion of the context for the work in the introduction is good. However, there are a few major issues related to 1) the findings and interpretation, 2) identification of initiator and follower events, and 3) controlling for context.

Major comments

1) findings and interpretation: My own interpretation of the data and findings is that Guinea fowl display attraction to conspecifics to maintain cohesion, and display sex-based leadership. Indeed, authors show male guineafowl have disproportionate influence on group movement, and that when the group is becoming dispersed (and specifically dividing into two subgroups) the birds in the smaller sub-group join the larger sub-group. Evidence for compromise or choose is weak (it is derived from analyses of movement patterns of dyads). The emphasis placed on each finding varies depending on the section of the manuscript (contrast “shared” and “majority rule” emphasis of the abstract with the “sex-based leadership” emphasis of the discussion). Also, the study is framed as a replication study (lots devoted to this in the introduction) but really it uses an established method for examining leading and following in another system.

2) identification of initiator and follower events: A change in distance of $>3.5\text{m}$ is between dyads is criterion for an event. This could represent a change of 3.5m for two birds that are next to each other, or 3.5m for two birds that are 50m apart. I would suggest the former may be meaningful, the latter not. So, I would like to see more justification for a) 3.5m is used beyond it being larger than GPS error, and less than used for baboons (5m) because the birds are small/have more cohesive groups, and b) how far the birds are from one another when the change in distance occurs.

I would therefore suggest authors investigate if/show if 3.5m represents something meaningful. For e.g., if 3.5m sits firmly below the average bird distance, then events are likely not meaningful movements. Also, authors should change (extend) the distance over which pulls are explored and present these findings. Are pulls occurring over larger distances where the birds are more spread out? At very least some sensitivity analyses is needed on these distances.

3) context: The data are gathered over two different years for the two groups (analyses are shown for the two groups), but with groups the authors have information on season, and group sizes/individuals tagged. It does not seem that these are accounted or controlled for in analyses (e.g. Table S1). How does number of tags, or proportion of group tagged effect probability of following, or some time interval? Are birds more likely to follow in a given month or season, or particular habitat?

Minor comments

4) Line 178. What about also testing absolute number of times followed, rather than success? “Success” measures how much one bird moves around relative to how often some other bird follows it. “Frequency” of pulls measures how often a bird’s movements are followed generally, and since $>80\%$ of birds recorded simultaneously this tells you their overall influence on other’s movements.

5) Line 200. Why would the dispersing or natal sex influence leadership? Needs more context. There is a lot of literature on sex-based leadership and natal sex and local knowledge etc.

6) Line 263. Models predict when agreement is low, they average, and high they choose one or the other direction. The authors examine agreement of two initiators only. This means there can never be an average, so does not test a key prediction of models.

- 7) Line 268. How does distance between initiators affect their likelihood of following? See main point 2.
- 8) Line 288. This is a replication of the pigeon pair study but does not consider group decision making. This considers cohesion between individuals within a moving group.
- 9) Line 312. This shows that groups become more cohesive when they start to drift apart into two subgroups. This will likely happen when birds are foraging at two food patches and then come together to move.
- 10) Methods: A look on google maps at the site suggests there are some clear habitat types. Where/when do the events extracted occur, and how does the spread of the group/decision events change over time/space?
- 11) Line 324-327. Related to point 1. "any individual could initiate movement, with no direct link between dominance and influence, confirming that vulturine guineafowl share decisions" this statement is misleading, and at odds with much of the rest of the discussion. Shared decisions does not mean no overall effect of dominance – it means all/most individuals can lead/direct group movement. This is not the case here. The authors show a clear sex-based leadership in both groups. This is important. In many species, both females and males more often lead groups. There is lots of evidence for male-biased leadership and female-based leadership. Female leaders are more likely to emerge within small egalitarian groups. This finding is of relevance for understanding how sex differences relate to leadership emergence and traits. Whilst this finding is discussed throughout the discussion (especially paragraph lines 341-353), the discussion contrasts with much of the rest of the write-up (including the start of the discussion). I think the sex-based leadership should be given more prominence in the title, abstract, etc.
- 12) Line 264-65. "For simplicity, in this test we focused on events comprising two initiators." Related to main issue number 1 and 2 raised above.
- 13) Line 504-519. What was the range and average duration of these events? How much of the events represent movements within a minute, or few seconds, etc.?
- 14) Supp material Table 1. The number of initiators effect size is negative; is this because of the interaction term fitted in the model? Authors should report the independent effects of agreement and initiators, in addition to the interaction.
- 15) Table 4. The data are gathered over different years for the two groups, and across seasons, with different group sizes/individuals tagged. It does not seem that these are accounted or controlled for in analyses (e.g. Table S1). How does number of tags, or proportion of group tagged effect probability of following, or some time interval? Are birds more likely to follow in a given month or season, or particular habitat?

Reviewers' comments:

Reviewer #1 (Remarks to the Author):

This manuscript is about the ways the members of the groups of terrestrial bird vulturine guineafowl (from now on birds) make compromises and choices while moving around. It focuses on several aspects arising in the above context. The authors consider the roles of such factors as dominance, the number of simultaneously acting (initiator) birds and other possible factors including, e.g., whether the collective motion is initiated by males or females. Another important aspect of their study is the comparison of their results with those determined for the collective movements of baboons and with those which are expected from a related model concerning leadership in group motion. The paper is well written, the results are interesting, however, I have a few points which I think should be clarified before the manuscript becomes suitable for publication. Before mentioning some of my concerns, I would like to point out that I agree with one of the main concepts of the work: I also think that demonstrating that the same behaviour occurs for different species is an important goal. Because of the complexity of animal societies, it can be instrumental if similarities or differences are found. In this sense I welcome “replication”.

We are grateful to Reviewer 1 for recognizing the significance of replication in science, particularly within the context of studying animal societies. We share the view in the paramount importance of replication, which is why we dedicated substantial resources and efforts to carry out the current study and hopefully our work will inspire other researchers in this direction. By exploring differences and similarities in collective decision-making processes, we aim to foster a deeper understanding of the underlying principles that govern these behaviours.

On the critical remarks side, I miss information/data regarding three relevant issues:

1) One of the main results (statement in the paper) is that the patterns of collective movement of the birds are analogous to those found for groups of baboons and predicted by theory. However, I do not find any quantitative comparison (figures, etc)

of the present data and those which they are compared with. In other words: is the agreement they find quantitative to any degree? Or is the behaviour “similar” only?

We added specific and quantitative comparisons in all subsections of the Results. Some examples:

Added in the first, section “Sex but not dominance determine who has influence”: ...
“The natal sex also has more information about the local landscape than the dispersing sex, which may contribute to observed differences (though we note that the baboon study did not explicitly tested for a sex difference). This difference may not, however, always play a role in decision-making. Females were often as successful at initiating as male group members, and many females initiated movements more often than some males. The relatively small differences between male and female vulturine guineafowl is likely to reflect the relatively low rates of conflict in most of their collective movements. We found that guineafowl are substantially more likely to follow initiators (P_{success} range: 0.7–0.9) than baboons (P_{success} range: 0.2–0.8)¹³. This is likely to explain the high degree of cohesion and small intra-group dispersion of vulturine guineafowl.”

In the legend of Figure 4 we already had a quantitative comparison between guineafowl and baboons but we now moved it to the results of the main text to better highlight this.

In the next section of the results “Followers compromise the initiation directions when initiators agree but choose a direction when initiators disagree”, we added a new paragraph that compares baboons to guineafowl (L330-340). In our last results section we highlight that baboons, similar to guineafowl follow a majority rule when in the choose regime but we also discuss some differences that our analyses revealed (L368-373 and 454-466).

2) The authors have already published several papers on these birds. It is not clear from the Methods section whether this study uses original data, or it uses data obtained earlier (for purposes involving movement interactions among the birds), and thus the present work being a re-analysis of prior data.

The vulturine guineafowl project is a long-term project that started in 2016 and it has been focusing on the study of the social organisation, collective decision-making and broader behavioural ecology of this species. A comprehensive outline of the study's design can be found in He et al. (2022), published in *Methods in Ecology and Evolution*.

This present manuscript utilizes the complete dataset of whole-group tracking of Group 1 and Group 2 that were collected for the very purpose of the present study. Through this, we aim to address questions pertaining to the mechanisms underlying collective decision-making. Our previous studies have employed selective portions of the data: In Papageorgiou et al. (2019) in *Current Biology*, we employed a subset of Group 2's data to derive specific social network thresholds, primarily relying on pairwise distances. Additionally, in Papageorgiou & Farine (2020) in *Science Advances*, another subset of Group 2's data was utilized to exemplify group cohesion based on pairwise distances. This subset also facilitated the calculation of parameters such as the distance from the group's centroid and the velocity of the final individual departing from the focal food patch, as presented in that particular paper. However, the overall dataset has never been analysed prior to the current study.

By using the complete group tracking data from both Group 1 and Group 2, our current work aims to enrich our understanding of collective decision-making processes within vulturine guineafowl and more broadly animal societies.

3) Since the whole work is about motion, I miss any visual demonstration of the processes described in the manuscript. I think that in 2023 a paper about the group motion of animals should have a few supplementary videos. I understand that it is kind of late to make videos of the birds (I certainly would have tried to do it, e.g., from above, using a distant, quiet drone, it takes only a few days that the birds would get completely used to it), but the authors have a huge amount of GPS coordinates versus time, so creating an animation of those would have not represented an enormous effort.

We acknowledge the importance of visualizing our data, and as a response to this, we have developed a series of three supplementary MP4 movies for each group (six in total). These videos effectively showcase the collective movement patterns derived from our GPS data. We also added an illustration on the top of the new Figure 1 showing what an initiation looks like.

Regarding the utilization of drones for enhanced visualization of guineafowl movement, we concur that drones could provide a more comprehensive perspective, even capturing social behaviours during tracking. However, integrating drones into our study would face significant challenges.

Primarily, the vulturine guineafowl face frequent risks from aerial predators such as martial and tawny eagles. This would likely present a hurdle in habituating them to any aerial presence nearby. It is worth noting that even if we were successful with habituating one group, the neighbouring non-habituated groups could be affected by drone-usage, although this aspect hasn't been explored extensively yet.

Moreover, our decision not to employ drones stems from the difficulties researchers have been encountering in obtaining research permits for drone usage in Kenya. The stringent regulations and authorization procedures compelled us to opt for a tracking methodology that circumvented potential permission-related issues.

In summary, while drones offer promising visualization possibilities, the current challenges of habituation and research permit limitations have led us to focus on GPS approaches that ensure the smooth progression of our research and provide long-term, high-resolution data.

I have a few – in part minor - comments which are likely to be helpful in making the manuscript clearer about some relevant details.

- Minor: In Fig. 1 the panel (iv) is not referred to (at least I do not see what is along the x axis)

x-axis label added.

- One of the most important aspects of the phenomenon is an event called “simultaneous pull”. Now, here the word simultaneous is essential and assumes a time scale which is much shorter than the typical time scale needed for a bird to make a distance compatible with the inter-bird distance. Simultaneous and “subsequent” are important aspects here and without explicitly specifying Δt , the time interval within which movements are considered to be simultaneous, the presentation of the data is incomplete.

In this context, it's important to clarify that Δt is not determined by a specific time threshold, but rather, it pertains to instances where pulls and anchors overlap temporally. To elaborate, the term "simultaneous" denotes initiation attempts (that will end up in either a pull or an anchor) with the same focal follower but different initiators. Two or more attempts are considered simultaneous if their initiation phases (increasing inter-individual distance) commence prior to the conclusion of any preceding initiation attempts. All such concurrent attempts are classified into a single **event**.

We now added explanatory sub-sentences of “simultaneous” in the Results and Methods sections. To provide enhanced clarity and visualization, we have included two supplementary figures. The first figure illustrates the distribution of initiation durations in minutes (Supplementary Figure 2A). The second figure portrays the distribution of event durations in minutes (Supplementary Figure 2B). These additions are intended to provide readers with a more comprehensive understanding of the temporal dynamics and durations within the observed pulls or anchors, and their associated events.

- Minor: I find that the following sentence is so complicated that it is difficult to make a clear conclusion out of it:

“We found that the number of simultaneous initiators, the level of their directional agreement (ranging from 0 when the movement vectors have an 180° difference, to 1 when the movement vectors have no difference), and their interaction predicted the probability of following a given initiation.”

We have now shortened the sentence and defined “directional agreement” in the previous paragraph (L271-273).

Reviewer #2 (Remarks to the Author):

“Compromise or choose: Shared movement decisions in wild vulturine guineafowl”

Summary and overall impression of the work

This paper uses high-quality trajectory data for wild guineafowl groups. The study system, site, and data are wonderful, and it is exciting to see such analyses being undertaken. Acquiring these data takes a huge effort and dedication – well done to the authors. The framing of the study and discussion of the context for the work in the introduction is good. However, there are a few major issues related to 1) the findings and interpretation, 2) identification of initiator and follower events, and 3) controlling for context.

Major comments

- 1) findings and interpretation: My own interpretation of the data and findings is that Guinea fowl display attraction to conspecifics to maintain cohesion, and display sex-based leadership. Indeed, authors show male guineafowl have disproportionate influence on group movement, and that when the group is becoming dispersed (and specifically dividing into two subgroups) the birds in the smaller sub-group join the larger sub-group. Evidence for compromise or choose is weak (it is derived from analyses of movement patterns of dyads). The emphasis placed on each finding varies depending on the section of the manuscript (contrast “shared” and “majority rule” emphasis of the abstract with the “sex-based leadership” emphasis of the discussion). Also, the study is framed as a replication study (lots devoted to this in the introduction) but really it uses an established method for examining leading and following in another system.

The reviewer raises four points, that we address below:

- (1) Vulturine guineafowl, like baboons, form highly cohesive groups. This means that 'becoming dispersed' is on the scale of tens of meters, as opposed to the kilometres that groups range over during the course of a day. Thus, separation between sub-groups—as noted by the reviewer—is quite distinct from species that express fission-fusion dynamics. Rather, 'dispersion' is likely to arise from temporary conflicts of interest in terms of directional preferences among group members, and these are resolved through collective decision-making (i.e. reaching consensus about where to move next). Thus, we perceive attraction, cohesion, and shared decision-making as intertwined concepts.
- (2) A key finding is that every individual group-member has the ability to initiate movement successfully, even if males tend to achieve slightly higher success rates (c. 10% differences in the probability of being followed, see also the new Figure 2). Consequently, we conclude that guineafowl employ a shared but graded rather than equal leadership system, as explained in our manuscript. The concept of decisions being shared is not synonymous with decisions being distributed entirely equally. Finally, leadership success is not only defined by the probability of being followed, but also by the number of attempts. There are therefore females in the group that are successful more often than males, simply because they attempt to initiate more often (see new Figure 2).
- (3) We respectfully disagree that evidence for choose versus compromise is weak. First, the patterns at the individual level necessarily scale up to the patterns at the group level, because the groups remain cohesive. Second, our findings align almost perfectly with theoretical models of collective decision-making in which this pattern is an emergent property. Finally, we statistically distinguish between the choose versus compromise regimes, confirming that these are robust patterns in the data. We note that the 'weak evidence' may stem from a misunderstanding of averaging by the reviewer (see our response to comment number 6 below).
- (4) We agree that our study is not a true replication, in that it is not conducted on baboons. However, we do replicate all of the methods from the baboon study, and also conduct a within-study replication (across two independent groups).

Further, our study was designed exactly to replicate the baboon study, rather than re-using existing data. We therefore believe that there is substantial utility in maintaining the framing of our study as a replication.

2) identification of initiator and follower events: A change in distance of $>3.5\text{m}$ is between dyads is criterion for an event. This could represent a change of 3.5m for two birds that are next to each other, or 3.5m for two birds that are 50m apart. I would suggest the former may be meaningful, the latter not. So, I would like to see more justification for a) 3.5m is used beyond it being larger than GPS error, and less than used for baboons (5m) because the birds are small/have more cohesive groups, and b) how far the birds are from one another when the change in distance occurs.

I would therefore suggest authors investigate if/show if 3.5m represents something meaningful. For e.g., if 3.5m sits firmly below the average bird distance, then events are likely not meaningful movements. Also, authors should change (extend) the distance over which pulls are explored and present these findings. Are pulls occurring over larger distances where the birds are more spread out? At very least some sensitivity analyses is needed on these distances.

In general, the Strandburg-Peshkin et al. 2015 methodology has been designed to avoid the use of thresholds. The 3.5 meters threshold in our study is the minimum distance we could choose that could be both biologically meaningful given the spatial cohesion of these birds, while also being safely larger than the GPS error and not excluding most of our data (see L565-569 and L578-584). We consider that the avoidance of other distance or time thresholds to characterize pulls, anchors or events as an important strength of this methodological approach.

The reviewer is correct, however, that the salience of a change in distance is substantially greater when the dyadic distance is short. To address this, we have followed the reviewer's advice of adding a new Figure in the main text (Figure 1) that shows the distributions of distances between leaders and followers during pulls and anchors at different times when they take place (i.e. when the leader starts moving—

t1, when the follower starts moving toward the leader or when the leader starts moving back—t2 and when the pull or anchor is completed—t3). We are very grateful for this suggestion, as this figure very clearly demonstrates the small spatial scale over which these pulls and anchors are taking place. It also argues for keeping a 3.5 meters threshold, as increasing the threshold would exclude most of our data (see text added in see L543-547 and L558-562) whereas decreasing the threshold would include too many noisy events.

3) context: The data are gathered over two different years for the two groups (analyses are shown for the two groups), but with groups the authors have information on season, and group sizes/individuals tagged. It does not seem that these are accounted or controlled for in analyses (e.g. Table S1). How does number of tags, or proportion of group tagged effect probability of following, or some time interval? Are birds more likely to follow in a given month or season, or particular habitat?

We agree that potential contextual effects are interesting, and investigating these was one of our original motivations in this study. However, two factors caused us not to include contextual factors in our study. The first is that our methods are extremely data hungry—they require tens of thousands of events in order to determine between the compromise and choose regimes. The limitation that this has is evident from our data on group 2 in the manuscript. The second is that we found almost no evidence for any variation caused by contextual factors. In her thesis, Danai (first author) explored whether being on a glade (open areas where guineafowl usually forage) versus in dense vegetation affects initiation success and the critical angles of the compromise and choose regimes. No difference emerged from this analysis.

As an additional note. When discussing the pros and cons of considering and/or including contextual factors in these results, we decided that any analyses are fraught with challenges. For example, we cannot say whether we have insufficient data or whether there is indeed no difference. Moreover, our classifications of habitat categories may not adequately capture the effect of context on collective decision-making.

Finally, none of our results appear to be sensitive to either group size or to the proportion of individuals tagged. First, given that the two groups studied exhibited more similarities than differences despite their differing sizes, we don't anticipate substantial intra-group size variation concerning decision-making processes. Second, our sensitivity analysis (outlined in Supplementary Analysis) demonstrates that our results remain robust to fluctuations in the proportion of tags operating during data collection.

Minor comments

4) Line 178. What about also testing absolute number of times followed, rather than success? "Success" measures how much one bird moves around relative to how often some other bird follows it. "Frequency" of pulls measures how often a bird's movements are followed generally, and since >80% of birds recorded simultaneously this tells you their overall influence on other's movements.

We have now added a new figure and permutation tests in the main text (see Figure 2) that address this very relevant and constructive Reviewer comment.

5) Line 200. Why would the dispersing or natal sex influence leadership? Needs more context. There is a lot of literature on sex-based leadership and natal sex and local knowledge etc.

We added a few explanatory sentences there (see lines L248-252).

6) Line 263. Models predict when agreement is low, they average, and high they choose one or the other direction. The authors examine agreement of two initiators only. This means there can never be an average, so does not test a key prediction of models.

The reviewer is misunderstanding the averaging here—when there are two initiators (potential pullers) with a relatively small angle, the follower averages their direction. This is the average of the two initiators, not the average of two followers.

7) Line 268. How does distance between initiators affect their likelihood of following?
See main point 2.

We added in the supplementary material two extra GEEs (see Supplementary Table 8), one per group, that predict the probability of following not only given the number of initiators, their level of agreement and their interaction. We also accounted there for the proportion of tags that were operating and the distance between the leader and the follower, when each pulling and anchoring took place. Even though these results matched very well our main GEE models, we found that when fewer tags operate it is more likely for an event to be detected as successful in Group 1 and less likely to be detected as successful in Group 2. Additionally, when the distance between the leader and the follower is larger (but no more than 30 meters, as we only consider cases where the group is cohesive, see Figure 1) the leader is more likely to be successful in both groups.

8) Line 288. This is a replication of the pigeon pair study but does not consider group decision making. This considers cohesion between individuals within a moving group.

Line 288 in the previous version of the manuscript corresponds to the legend of Figure 5, which replicates Figure 3B from the baboon study by Strandburg-Peshkin et al. (2015, Science). The referenced baboon study is on group decision-making. Our intention in dividing this section, where this figure is presented, is to now enhance the clarity of the transition from the two-puller context to the majority rule. By doing so, we aim to improve the persuasiveness of our manuscript in regard to its contribution to the field of group decision-making/ collective behaviour.

9) Line 312. This shows that groups become more cohesive when they start to drift apart into two subgroups. This will likely happen when birds are foraging at two food patches and then come together to move.

We hope that our explanations above better clarify this part of the manuscript.

10) Methods: A look on google maps at the site suggests there are some clear habitat types. Where/when do the events extracted occur, and how does the spread of the group/decision events change over time/space?

Please see above our response on contextual factors concerning the physical and social environment. We also added one sentence in the discussion (L450-452): *“The potential influence of the social, as well as the physical environment is worth exploring, given that environmental effects have already been documented across various facets of collective behavior”*

11) Line 324-327. Related to point 1. “any individual could initiate movement, with no direct link between dominance and influence, confirming that vulturine guineafowl share decisions” this statement is misleading, and at odds with much of the rest of the discussion. Shared decisions does not mean no overall effect of dominance – it means all/most individuals can lead/direct group movement. This is not the case here. The authors show a clear sex-based leadership in both groups. This is important. In many species, both females and males more often lead groups. There is lots of evidence for male-biased leadership and female-based leadership. Female leaders are more likely to emerge within small egalitarian groups. This finding is of relevance for understanding how sex differences relate to leadership emergence and traits. Whilst this finding is discussed throughout the discussion (especially paragraph lines 341-353), the discussion contrasts with much of the rest of the write-up (including the start of the discussion). I think the sex-based leadership should be given more prominence in the title, abstract, etc.

Please see above our responses and changes. Also we note that we specifically found no effect of dominance on leadership, only an effect of sex. Further, we re-emphasise that shared decision-making does not mean that all individuals actually have equal influence, only that they can have equal influence (differences may arise based on motivation, experience, etc.).

We have, however, rephrased sentences in the first paragraph of our discussion as follows: *“In both our guineafowl study groups, we found that any individual could initiate movement, with no direct link between dominance and influence. Male*

guineafowl are more likely to be followed than females, and also have slightly higher rates of initiations. However, females still initiated often, and many had a high number of successful attempts..”

We also added graded leadership in our keywords and ensured that the finding regarding males remains in our abstract after shortening it to be under the word limit.

12) Line 264-65. “For simplicity, in this test we focused on events comprising two initiators.” Related to main issue number 1 and 2 raised above.

See responses above.

13) Line 504-519. What was the range and average duration of these events? How much of the events represent movements within a minute, or few seconds, etc.?

We have now added Supplementary Figure 2 that gives all this information.

14) Supp material Table 1. The number of initiators effect size is negative; is this because of the interaction term fitted in the model? Authors should report the independent effects of agreement and initiators, in addition to the interaction.

There is debate in the literature as to whether coefficients can be interpreted independently when there is a significant interaction. For example, while having more initiators has a positive effect on followership when agreement is high, it has a negative effect when agreement is low. Thus, the sign of the coefficients cannot be interpreted independently of the interaction term. For this reason, we plot the model predictions (Figure 4), as this makes interpretation much simpler. We also had already stated this flipping in the effect of the number of initiators in the main text (L279-284).

15) Table 4. The data are gathered over different years for the two groups, and across seasons, with different group sizes/individuals tagged. It does not seem that these are accounted or controlled for in analyses (e.g. Table S1). How does number of tags, or proportion of group tagged effect probability of following, or some time

interval? Are birds more likely to follow in a given month or season, or particular habitat?

See more on the aspect of seasonality and habitat context above. For all the analyses we have run we have set the rule that the individuals involved as leaders in an event, as well as the follower, to be within 30 meters from the group's centroid at the time when the event takes place, as within-group pairwise distances are below 30 meters at 95% of the time in our study species (Papageorgiou et al. 2019, *Current Biology*). We are thus confident that this threshold captures only cases when our study groups are cohesive and therefore, even if seasonality impacts group's cohesion—which is not the focus of the current study but a potentially interesting question—it may not impact the decision-making processes we investigate here. In any case, as we can see in the new Figure 1 that we just added, leaders are rarely more than 10 meters from the focal follower when they start an initiation and therefore the threshold of 30 meters excludes only a very small number of events. See more detailed responses to earlier comments above.

REVIEWERS' COMMENTS:

Reviewer #2 (Remarks to the Author):

I find that my remarks have been satisfactorily addressed in the revised version of the MS.

Reviewer #4 (Remarks to the Author):

I have read through the authors' revised manuscript (including tracked changes) as well as the reviewers' comments and the authors' responses to these comments, with a focus on their response to Reviewer #2. I have specifically been requested to assess the authors' responses to Reviewer 2. I appreciated this manuscript and think that it makes an important contribution by replicating specific methods applied to the study of baboons in a new species, vulturine guineafowl, to better understand similarities and differences in the behavioral mechanisms underlying collective behaviors across these species as well as across two different groups of the same species.

Overall, I believe that the authors have adequately addressed the points raised by Reviewer #2. One of the most substantial points raised by Reviewer #2, and which was also emphasized by the editor, was the selection of the 3.5m threshold for identifying initiation attempts (i.e., an initiator needs to move at least 3.5 meters away from the other member of the dyad for the authors to include the initiation attempt in analysis). They question the rationale for this threshold and request more data on how far away the birds tend to be when initiation attempts are made. They also request sensitivity analyses regarding this threshold.

The authors argue that the threshold must be greater than the error in the GPS points (1 meter) but not too large so as to discard a large portion of their data. The authors responded by adding a new figure, Figure 1, which presents the distribution of distances between dyads that engaged in pulls or anchors at the beginning and end of an initiation and the end of a pull or anchor. These data are strongly right-skewed and show that dyads tend to have very close distances between them at the beginning of an initiation and the end of a pull or anchor. When considering the distribution of distances at the end of an initiation attempt, a large percentage of the distances fall less than 20 meters. Thus, they argue that a threshold of 3.5 meters is a meaningful change in distance. The authors use these results to justify their choice of the 3.5-meter threshold, arguing that 3.5 meters is substantially higher than the typical distance between a pair of individuals while low enough to allow the capture of most initiation attempts. I think that the addition of these data is very useful for assessing the relevance of this threshold. Although I think that sensitivity analyses would be informative regarding how results might change, for instance, when only considering longer initiation attempts, I am satisfied that this choice of threshold is reasonable, given the data provided. In addition, in response to another comment by Reviewer 2, the authors have also added a Supplemental analysis that considers how the distance between the leader and follower impacts the likelihood of a successful initiation.

An additional comment of Reviewer 2 that both this reviewer and the editor thought would be important to address relates to controlling for potential differences between the two groups of birds in the analyses, for example, by exploring the role of the ecological environment, group size, and also the proportion of tags that are functioning. The authors point to Supplemental analyses that indicate that the proportion of individuals tagged do not affect the results. In particular, they replicate all study results after only considering instances when at least 80% of the tags are functioning. Furthermore, they point out that although the two groups did differ in group size, they found very similar results. As for controlling for the ecological context across these two groups, the authors agree that this is an

interesting question. However, they point to analyses presented in the lead author's dissertation indicating that the openness of the habitat was not related to the success of initiations or the critical angles differentiating between regimes. My own perspective is that while the effect of the surrounding habitat is an interesting question, taking it into account is not necessary for the analyses presented in this manuscript. This is especially the case because the authors are also comparing the results to the study of baboons which also took place during a different time period. As the aim of the study is to determine the generality of the results across species, I think that the results are actually quite robust since they hold up across quite different contexts.

Reviewer 2 also stated that evidence for compromise vs. choose was weak. The authors respectfully disagree with this assessment and suggest that the reviewer may have misunderstood the analysis. Indeed I believe there was a misunderstanding regarding who was traveling in the average direction, the leaders or the follower. The authors point out that it is the followers that travel in the mean initiation direction of the initiators (as long as the disagreement between their initiation angles is below a given threshold). Thus, I agree with the authors that the study results do test the specific prediction of the model.

The authors also made some helpful changes as a result of some of the Reviewer 2's other minor comments, such as examining both the success of initiations and their frequency. They also clarified the nature of the shared leadership that occurs in this group, making it clear that while all individuals can, and do, initiate successfully, males do so more than females.

Overall, I think that Reviewer #2's comments raised some important points. Based on my assessment, I think that the authors have sufficiently addressed these points, resulting in an improved manuscript that I support accepting for publication.

Dr. Lisa O'Bryan
Rice University